# Improving topic modeling performance on social media through semantic relationships within biomedical terminology

Yi Xin[1,2], Monika E. Grabowska[2], Srushti Gangireddy[2], Matthew S. Krantz[2,3],
V. Eric Kerchberger[2,3], Alyson L. Dickson[3], Qiping Feng[2,3], Zhijun Yin[1,2], Wei-Qi Wei[1,2]*

**1** Department of Computer Science, Vanderbilt University, Nashville, Tennessee, United States of America,
**2** Department of Biomedical Informatics, Vanderbilt University Medical Center, Nashville, Tennessee,
United States of America, **3** Department of Medicine, Vanderbilt University Medical Center, Nashville,
Tennessee, United States of America

\* wei-qi.wei@vumc.org

## Abstract

Topic modeling utilizes unsupervised machine learning to detect underlying themes within texts and has been deployed routinely to analyze social media for insights into healthcare issues. However, the inherent messiness of social media hinders the full realization of this technique's potential. As such, we hypothesized that restricting medical concepts in social media texts to specific related semantic types and applying topic modeling to these concepts could be a feasible approach to overcome the challenge of traditional topic modeling for social media texts. Therefore, we developed a semantic-type-based topic modeling pipeline to discover self-reported health-related topics. This pipeline integrated semantic type information and Systematized Medical Nomenclature for Medicine (SNOMED) precoordinated expressions into a traditional topic modeling approach to enhance effectiveness in clustering meaningful, distinct topics. Using social media texts regarding statins for illustration, we evaluated the efficacy of this new approach and validated a newly identified topic using real-world clinical data. Based on expert evaluations, this approach resulted in more novel, distinguishable, and meaningful health-related topics compared to traditional topic modeling. In addition, our electronic health record validation for a newly identified topic in two real-world clinical databases indicated that statin users had a higher prevalence of depression or anxiety compared to matched non-users. Our results indicate that this new topic modeling pipeline can improve the extraction of themes from noisy online discussions, thereby contributing to deeper insights for healthcare research.

## Introduction

Topic modeling, an unsupervised machine learning technique in natural language processing (NLP), is used to discover thematic structures in large text collections [1]. It enables a wide range of text-mining tasks in healthcare (e.g., feedback analysis, clinical decision support, and research literature management), helping to extract themes, uncover latent relationships,

**Data availability statement:** All codes are publicly available on GitHub. (https://github.com/The-Wei-Lab/topic_model_semantic_type) The social media data from Reddit was held in an online web-scraped Reddit file repository at https://academictorrents.com/collection/datasetreddit. The clinical data for validation was obtained from All of Us Research Program (https://allofus.nih.gov/) and VUMC BioVU Program (https://victr.vumc.org/). The BioVU clinical summary data are available upon request.

**Funding:** WQW received the funding award. This study was funded by National Institutes of Health (NIH) research grant R01GM139891, R01AG069900, R01LM012806, R01HG012748, and R01HL163854. No sponsor or funder played any role in the study or preparation of the manuscript.

**Competing interests:** The authors have declared that no competing interests exist.

and enhance text understanding [2–4]. The structural topic model (STM) is one of the most common types of topic modeling approaches used widely in analysis of healthcare data and social media health discussions, as it excels in uncovering latent themes and temporal trends [5,6]. Unlike other topic modeling approaches, such as Latent Dirichlet Allocation, STM is uniquely able to incorporate document-level metadata, such as user attributes and temporal information, making it particularly advantageous for analyzing social media data in health-related NLP tasks [7–9].

Topic modeling has become an effective and prevalent approach to mining self-reported health information from online health-related discussions on social media platforms. For example, Chen *et al.* [10] conducted topic modeling and regression analyses to analyze cancer-related discussions on social media, evaluating the connections between various cancer topics and user engagement. Jo *et al.* [11] used STM and network analysis to examine public concerns over the early stages of the COVID-19 outbreak from an online Q&A forum. Likewise, Xin *et al.* [12] employed STM to uncover the themes and concerns from online discussions of rheumatoid arthritis patients on social media before and after the COVID-19 pandemic. Such insights gained from topic modeling have demonstrated its remarkable potential for detecting underlying themes from large-scale health-related discussions on social media.

While biomedical information extractions from social media texts can offer important insights to researchers, there are limitations and challenges associated with this type of topic modeling. Since individuals frequently use informal and inconsistent words to discuss their health experiences on social media, the relevant texts typically exhibit a variety of noise, including irrelevant content, misinformation, and emotional language [13,14]. Consequently, traditional topic modeling approaches are limited in their abilities to precisely extract health-related themes from a messy and unstructured text corpus on social media compared to structured clinical data [7]. Today, biomedical terminologies are widely used to formalize electronic health record (EHR) data and enable large-scale analyses. For instance, the Unified Medical Language System (UMLS), maintained by the National Library of Medicine, serves as a terminology repository for the most commonly used controlled vocabularies in the biomedical sciences [15]. It categorizes biomedical concepts into hundreds of semantic types (e.g., syndrome/disease and sign/symptom), providing a comprehensive framework for broadly classifying the biomedical domain. To address the problem of noise, studies have shown that restricting certain semantic types of words in social media texts can more efficiently reveal biomedical information. Rai *et al.* [16] restricted semantic types in social media texts to reveal how race moderates associations between depression and first-person pronouns or negative emotion words. Ru *et al.* [17] mapped diseases and symptoms mentioned in social media texts into standardized UMLS terminologies to enhance the accuracy of machine learning models in detecting serendipitous drug usage. Therefore, we hypothesize that restricting medical concepts in social media texts to specific related semantic types and applying topic modeling to these concepts could be a feasible approach to overcome the challenge of traditional topic modeling for social media texts.

Here, we assess statin-use discussions on social media as an example to illustrate the effectiveness of this new approach. We develop a semantic-type-based topic modeling pipeline to discover self-reported health-related topics on social media. This pipeline integrates semantic-type-based UMLS concept recognition and concept decomposition with Systematized Medical Nomenclature for Medicine (SNOMED) precoordinated expressions with the aim of improving the model's performance in clustering more novel, distinguishable, and meaningful biomedical topics, as assessed by content experts. We further validate

the legitimacy of any newly identified topic using real-world clinical data from two large-scale databases.

## Methods

### Pipeline overview

In this study, we began by collecting and preprocessing social media data extracted from Reddit. We then conducted an initial round of STM using a traditional STM pipeline. Next, we enhanced this pipeline by incorporating semantic type information through UMLS concept recognition using MetaMap and performed a second round of STM. Subsequently, we further extended the pipeline by integrating SNOMED precoordinated expressions through concept decomposition, followed by a final round of STM. We then compared the results from the second and final STM rounds. Finally, we validated a newly identified topic from the final round of STM to demonstrate the effectiveness of our approach.

### Data collection

We conducted web scraping of a repository containing dump files to collect the submissions and comments from the subreddit *r/Cholesterol* from January 1, 2017, to December 31, 2022 [18]. In terms of ethical considerations of using social media data for healthcare research, we ensured adherence to ethical standards and for data collection from social media platforms [19,20]. Specifically, the extracted social media data was de-identified and no personal characteristics from users were collected. The data resource has been described in section I, S2 Text. Fig 1 depicts the count of total submissions each year from 2017 to 2022. Specifically, we extracted submission creation time, comment count, score (the difference between upvotes and downvotes), title, and body text for each submission. We then filtered out submissions originating from deleted users, moderators, and prolific users with an abundance of irrelevant content. Furthermore, to identify submissions discussing any single-ingredient statin, we compiled a list of both generic and brand names, including 'statin', 'atorvastatin/lipitor', 'simvastatin/zocor', 'rosuvastatin/crestor', 'pitavastatin/livalo', 'fluvastatin/lescol', 'lovastatin/mevacor', and 'pravastatin/pravachol'. We then extracted submissions containing lowercase and singular forms of words that matched any name on the list.

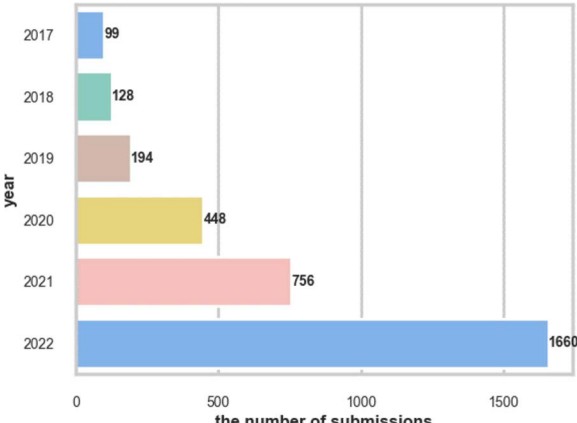

**Fig 1. The number of total submissions in *r/cholesterol* each year from 2017 to 2022.**

## UMLS concept recognition with MetaMap

We first preprocessed the collected data (section II, S2 Text). Then we used MetaMap (2023AA USAbase Strict Data Model) to identify UMLS concepts in each submission [21]. We restricted the UMLS Source Vocabularies to ICD-10-CM, ICD-9-CM, RxNorm, SNOMED CT United States, and used concepts under semantic types: "sosy" (sign or symptom), "dsyn" (disease or syndrome), "mobd" (mental or behavioral dysfunction), and "bpoc" (body part, organ, or organ component), as these semantic types are central to the health narratives shared in this online community. In our MetaMap implementation, we set the number of composite phrases to 4 and the prune threshold to 20 [21]. We then generated Field MMI (MetaMap Indexing) output to extract mapped CUIs in a concise and unified form of output text. By focusing on these semantic types, we ensured the extraction of clinically meaningful information while filtering out less relevant terms, improving the precision and relevance of topic modeling.

## STM model setup

The framework and parameters of STM have been described in section IV, S2 Text. STM generates two key probability distributions: the topic-document distribution ($\theta$) and the word-topic distribution ($\beta$). The topic-document distribution ($\theta$) represents the probability of each topic within a document, while the word-topic distribution ($\beta$) represents the probability of each word within a topic. These distributions are integral to further analysis. In addition, we performed an in-scope selection of the number of topics ($k$) for STM. Particularly, we assessed the model's performance in terms of semantic coherence and exclusivity. Semantic coherence measures the frequency of co-occurrence of words within a given topic in existing documents, while exclusivity quantifies the likelihood of words occurring exclusively within a topic with high probability [22,23]. In addition, the selection of the optimal number of $k$ is also influenced by the intrinsic nature of the corpus [24–26]. In the context of studies such as this one, before UMLS concept recognition, the number of topics ($k$) is typically relatively large due to the abundance of underlying themes in discussions. However, after UMLS concept recognition, the number of topics ($k$) is smaller because the corpus composed of mapped CUIs is relatively homogeneous due to the limited number of tokens and underlying themes for statin users in online communities, while larger $k$ values led to fragmentation [24–26]. Therefore, to determine the optimal number of $k$ after UMLS concept recognition, we generated visualizations of mean semantic coherence and exclusivity for models with $k$ values ranging from 2 to 6 before and after concept decomposition. Subsequently, we chose the number of topics that struck the best balance between the semantic coherence and exclusivity. In addition to the quantitative analysis, we conducted a qualitative assessment to evaluate the coherence and interpretability of the topics. For each topic, we examined the top CUIs with the highest probabilities, assessing their semantic and contextual alignment within the topic's overall theme. Furthermore, we reviewed the top 5 original documents associated with each topic to verify whether the extracted CUIs accurately represented the content and context of the documents. This approach ensured that the topics were not only statistically robust but also meaningful and distinct in their interpretations.

## Concept decomposition based on SNOMED CT relationship

To improve the effectiveness of topic modeling (i.e., to provide more meaningful and distinguishable clustering of topics from clinical insights and to filter out redundant or irrelevant UMLS concepts in documents), we decomposed specific UMLS concepts based on the SNOMED CT hierarchy. For instance, CUI C0231528 (myalgia) was decomposed into CUI

C4083049 (muscle) and CUI C0030193 (pain). To perform this concept decomposition, we first identified UMLS concepts with significant word-topic distribution $\beta$ values ($\beta > 0.02$) in each topic (e.g., myalgia) and compiled these into a list. Then, we searched each of the UMLS concepts from this list in SNOMED CT and examined whether it had precoordinated expressions, which meant that this medical concept was predefined and had a formal logic definition represented by a set of defining relationships to other concepts in SNOMED CT [27,28]. Next, based on the SNOMED parent-child relationships we found (e.g., muscle and pain are parent concepts for myalgia), we decomposed the CUI of each identified UMLS concept into the CUIs of its parent UMLS concepts within corresponding documents. Finally, we applied STM to the documents after performing the concept decomposition for identified UMLS concepts.

## STM model comparisons

We conducted a comparative analysis to evaluate the impact of concept decomposition on the performance of topic modeling. First, we compared the histograms for the highest 20% values of topic-document distribution ($\theta$) in each topic before and after concept decomposition. Next, we visualized the document relationships within each topic before and after the decomposition. To facilitate this analysis, we converted the document-term matrix into a TF-IDF matrix. Given the high dimensionality of the TF-IDF matrix, we applied principal component analysis (PCA) [29] to reduce its dimensionality and improve computational efficiency using the R package prcomp (version 3.6.2) [30]. Additionally, we calculated the proportion of variance explained by each component, determined the number of components needed to retain 95% of the variance, and then retained only those components in the TF-IDF matrix. Finally, we applied t-distributed stochastic neighbor embedding (t-SNE) [31] using the R package tsne (version 0.1.3.1) [32] to project the reduced TF-IDF matrix onto a two-dimensional space, thereby providing a visual representation of the data. At last, we performed a blind expert review to compare the two groups of topics (section III, S2 Text).

## EHR validation

We conducted two case-control studies using de-identified EHR data from Vanderbilt University Medical Center's (VUMC) and the National Institutes of Health All of Us Research Program to investigate the association between statin exposure and mental health conditions (i.e., depression and anxiety) [33,34]. We initially conducted a search for Phecodes related to depression and anxiety, mapped these to the International Classification of Diseases (ICD) codes and compiled them into a list [35,36]. The case group was comprised of adults (age ≥ 18 years old) who met the following criteria: (1) reported race as either Black/African American or White/Caucasian; (2) gender as either male or female; (3) exposure to any statin in our predefined list, as detailed in Data Collection. The control group included adults: (1) reported race as either Black/African American or While/Caucasian; (2) gender as either male or female; (3) no exposure to any statin in our predefined list. For all patients, we extracted the age at the last EHR visit, the duration of EHR records (calculated as the last visit year minus the first visit year), race, and gender. To ensure a balanced comparison, we matched controls to cases in a 1:1 ratio based on age, race, EHR duration, and gender. We then calculated the prevalence of depression in both groups by dividing the number of patients with a depression diagnosis by the total number of patients in each cohort. Finally, we conducted a two-tailed z-test to see if the difference of the two proportions was statistically significant (significance level = 0.05).

## Results

### Topic summary for documents

We retrieved documents from Reddit (n = 1085 after data collection and preprocessing, from January 1, 2017, to December 31, 2022) and then applied STM to extract themes from the original documents. Fig 2 shows the 11 initial topics identified by using STM, with their high-frequency words and topic proportions. Topic proportion is the percentage of the documents associated with a specific topic, indicating its prevalence. The most prevalent topic in this online community was topic 1, where patients communicated their side effects of taking statins and sought statin success stories. The second most prevalent topic was topic 2, where patients discussed their test results for cholesterol and making decisions for statin use. The third most prevalent topic was topic 3, where patients with high cholesterol looked for lifestyle change recommendations, such as diet and exercise, to ameliorate their cholesterol levels. Beyond these particularly relevant results (e.g., specific medicines, symptoms, and diseases), the STM approach also extracted many topic noises, such as "want", "take", and "week". In addition, it was difficult to fully differentiate between several topics, including topics 2 and 8 as well as topics 3 and 7, due to the overlapped topic noises. These ambiguous results left room for improvement for effectively extracting health-related topics from the original documents. Since semantic types can categorize words or phrases based on their meanings and roles within sentences to enhance understanding and processing of language data [37], we integrated semantic relationships into the traditional STM.

As a first step in enacting our pipeline, we applied UMLS concept recognition based on Metamap [21] to the original documents by restricting the semantic types to: "sosy" (sign or

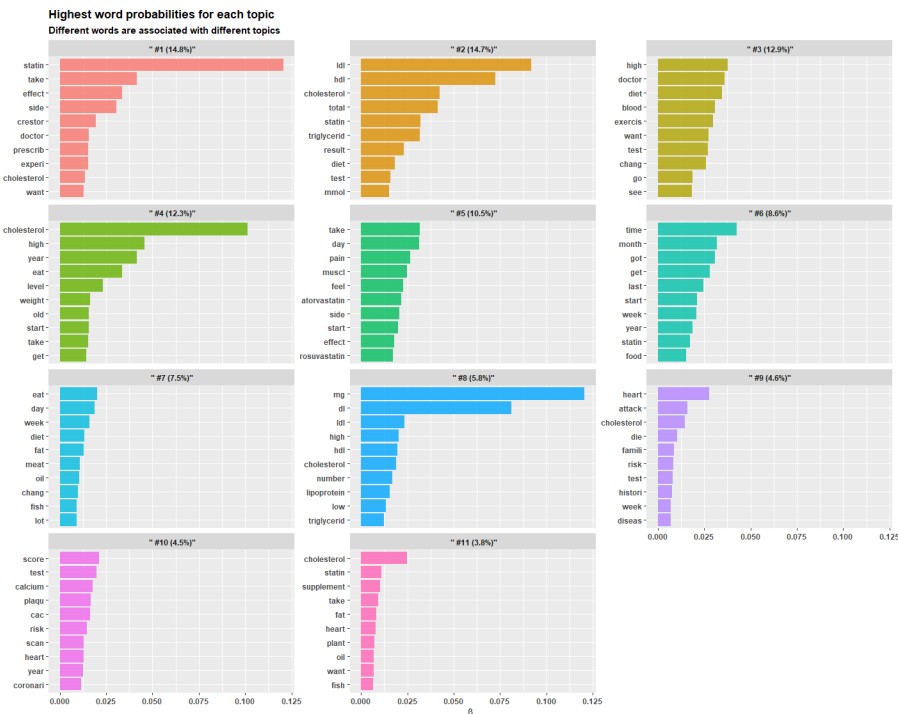

**Fig 2. The 11 mutually exclusive topics identified by using STM, in order of proportion.** Each subfigure is labeled with the topic number and its corresponding proportion. The bar plot visually represents the top 10 words with the highest probability for each topic.

symptom), "dsyn" (disease or syndrome), "mobd" (mental or behavioral dysfunction), and "bpoc" (body part, organ, or organ component). This restriction yielded 756 total documents, which were composed of Concept Unique Identifiers (CUIs) after the UMLS concept recognition, since some documents (n = 329) did not have the words from the restricted semantic types. For instance, documents asking general questions like "Atorvastatin 20 Mg and Side Effects?" and "Long term statin usage-how risky is it? I will likely be on it the rest of my life which will be another hopefully 40 years" were excluded. Since the number of topics ($k$) is an important parameter that needs to be determined by researchers before running STM, we evaluated the mean semantic coherence and exclusivity for models with UMLS concept recognition when varying the number of topics ($k$) before and after concept decomposition. Fig 3 presents the visualizations of mean semantic coherence and exclusivity for $k$ values ranging from 2 to 6 before and after concept decomposition. In addition, we reviewed the high-frequency UMLS concepts and associated original documents in each topic for models with $k$ values ranging from 2 to 6 before and after concept decomposition. Based on these quantitative and qualitative assessments, we chose the optimal number of topics $k = 3$ before and after concept decomposition, given the balance of semantic coherence, exclusivity, and interpretability, particularly in the context of clinical insights.

Fig 4(a) illustrates the three refined topics identified by STM before concept decomposition, detailing the top 10 UMLS concepts with the highest word-topic distribution ($\beta$) for each topic. Expert reviews observed that the topics were not aligned with clinical insights and lacked coherence. For example, some closely related UMLS concepts were separated into different topics, such as 'heart' in topic 2 and 'heart disease' in topic 3, creating overlap in the concepts represented by the topics and making it difficult to interpret their clinical relevance (i.e., both topic 2 and topic 3 contained similar UMLS concepts related to cardiovascular disease). In addition, the UMLS concepts in topic 1 captured aspects of metabolic syndromes, mental diseases, and side effects, which covered a large proportion of documents. To address this misalignment, we then conducted concept decomposition and determined UMLS concepts to be decomposed. We searched UMLS concepts with significant word-topic distribution $\beta$ values ($\beta > 0.02$) and identified CUI C0231528 (myalgia) and CUI C0002962 (Angina Pectoris) based on SNOMED precoordinated expressions. We then decomposed CUI C0231528 (myalgia) into CUI C0030193 (pain) and CUI C4083049 (muscle), and decomposed CUI C0002962 (Angina Pectoris) into CUI C0008031 (chest pain) and CUI C0018799 (heart

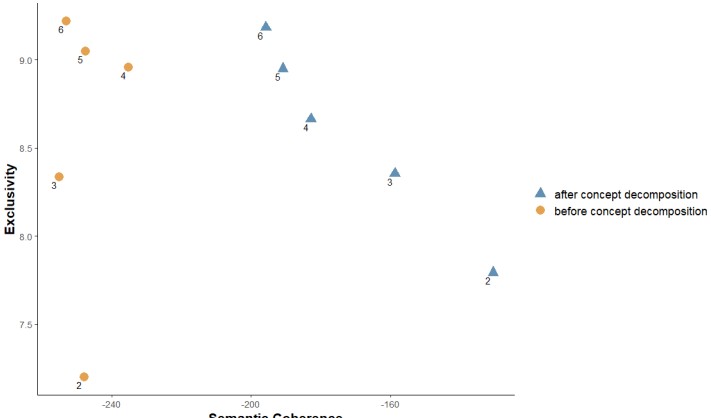

**Fig 3. Mean of the semantic coherence and exclusivity for models with UMLS concept recognition for k values ranging from 2 to 6 before and after concept decomposition.**

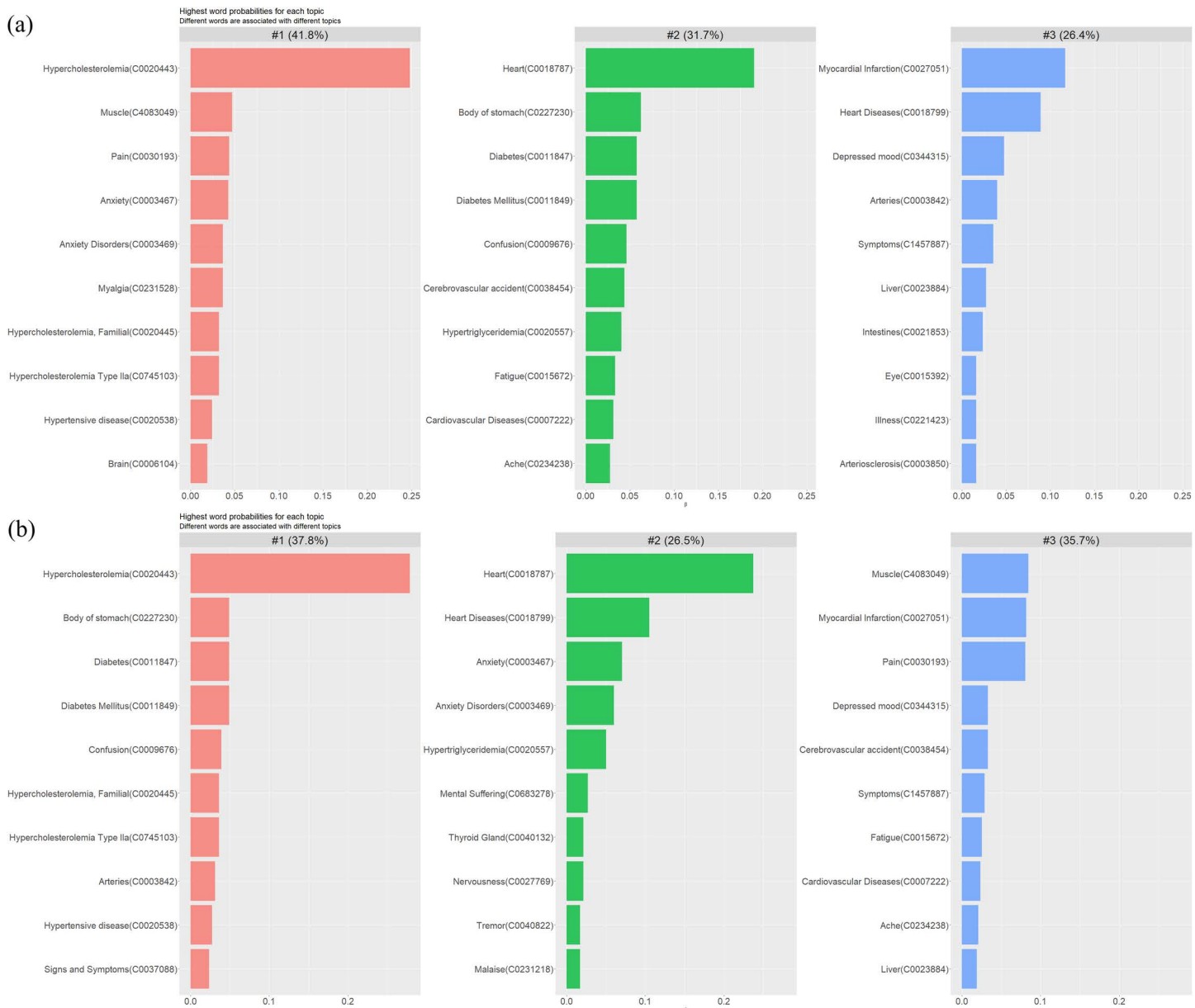

**Fig 4. The 3 topics identified by using STM (a) before and (b) after concept decomposition.** Each subfigure is labeled with the topic number and its corresponding proportion. The bar plot visually represents the top 10 UMLS concepts with the highest probability for each topic.

diseases) based on SNOMED parent-child relationships within the corresponding documents and reran STM on these documents following the concept decomposition.

Fig 4(b) presents the three optimized topics identified by STM after concept decomposition along with corresponding proportions. We observed that topic 1 was the predominant topic after concept decomposition, holding the majority at 37.8%, as shown in Fig 4(b). This topic contained high-frequency UMLS concepts related to metabolic abnormalities, such as "hypercholesterolemia", "diabetes", "diabetes mellitus", and "hypercholesterolemia, familial". Closely following was topic 3, which also held a substantial share (35.7%). This topic included high-frequency UMLS concepts such as "muscle",

"myocardial infarction", "pain", and "fatigue". The third topic was topic 2 (26.5%), characterized by high-frequency UMLS concepts such as "heart", "anxiety", "anxiety disorders", "mental suffering", and "nervousness". Based on expert evaluation, the group of topics after concept decomposition in Fig 4(b) appeared to have a more clinically coherent clustering. These topics were more specifically focused on well-recognized clinical syndromes or symptoms (i.e., metabolic syndrome, anxiety and depression, muscle pain), which is more beneficial from a clinical perspective. Each topic in this group captured a distinct aspect of health, aligning well with how health conditions are categorized and treated (section III, S2 Text). The example original documents associated with each topic in Fig 4(b) are attached in S3 Text.

### Topic-document probability distribution comparisons

Fig 5 depicts the comparisons of topic-document probability ($\theta$) histograms for the highest 20% values in topic 1, topic 2, and topic 3 with concept decomposition before and after concept decomposition. Notably, a significant increase can be observed in the values of the highest 20% probabilities for all three topics following concept decomposition. Additionally, we compared the number of associated documents for each topic before and after concept

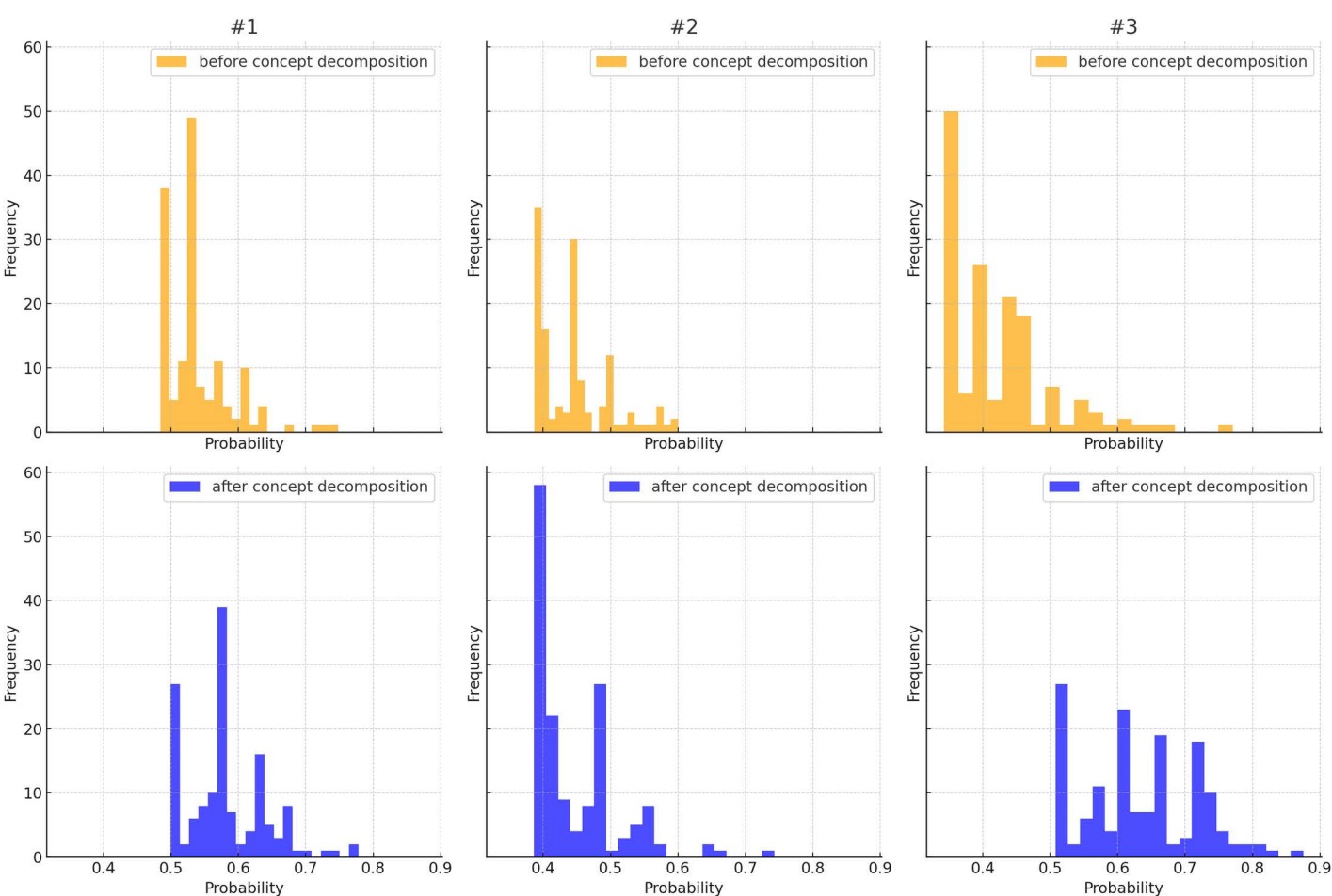

**Fig 5. Topic-document probability histograms for the highest 20% values in topic 1, topic 2, and topic 3 applied UMLS concept recognition before and after concept decomposition.**

decomposition. For topic 1, the count of associated documents decreased from 485 to 314, accompanied by a substantial rise in the number of documents with a probability exceeding 0.6. In the case of topic 2, the document count stayed the same (169), but with a concurrent increase in the number of associated documents exceeding a probability of 0.6. Topic 3 demonstrated the most pronounced changes, with the associated document count increasing from 102 to 273 and a substantive increase in documents with a probability between 0.6 and 0.9. The histograms of topic probability distribution indicate that documents were more likely to be statistically significant and distinguishable among the 3 topics following concept decomposition.

## Visualizations of topic-document relationships

Fig 6 displays the results of t-SNE visualizations of the TF-IDF matrix for our dataset, where each point represents a document, and the color indicates the document's primary topic. Before concept decomposition, some dense clusters were formed by points representing multiple topics. However, after concept decomposition, points corresponding to the same topic were positioned in relatively closer proximity to each other, rather than being clustered with points from other topics. In particular, certain groups of points tended to be dominated by a single topic. For instance, a large set of points in the center belonged to topic 1, which is a prominent topic in our documents. Furthermore, points associated with topic 3, were grouped into more distinct clusters, becoming more distinguishable from points associated with other topics. Therefore, the topic-document relationships became more clearly defined following concept decomposition.

## EHR validation

Since we newly identified topic 2 related to anxiety and depression, we conducted two EHR-based case-control studies using clinical data from Vanderbilt University Medical Center (VUMC) and the National Institute of Health All of Us Research Program to compare the prevalence of depression between statin users and matched statin non-users. We first identified statin users based on our predefined statin list in Data Collection and then identified patients with depression or anxiety using ICD-9 and ICD-10 codes mapped from relevant Phecodes (Table 1 and Table 2 in S1 Text). Patients with any of these codes in their visit records were considered to have experienced depression or anxiety. Table 1 indicates the prevalence of anxiety/depression among statin users and non-users using EHR data from VUMC and All of Us. In each of the two clinical databases, we found that the rate of depression among statin users was significantly higher than that among matched non-users (p < 0.00001 in both cohorts).

## Discussion

This study demonstrates that the integration of UMLS concept recognition and concept decomposition based on SNOMED CT relationships into a traditional topic modeling framework can enhance the definition of meaningful biomedical topics. Traditional token-based

**Table 1. EHR validation for anxiety/depression of statin-use patients using data from VUMC and All of Us.**

|  | Case: Statin users, n | Control Pool | Matched control, n | Depression in case, n (%) | Depression in matched control, n (%) | P-value of z-test |
|---|---|---|---|---|---|---|
| VUMC | 251,271 | 1,621,046 | 251,271 | 62,301 (24.79%) | 43,911 (17.48%) | <0.00001 |
| All of Us | 10,409 | 234,772 | 10,409 | 5,264 (50.57%) | 4,490 (43.14%) | <0.00001 |

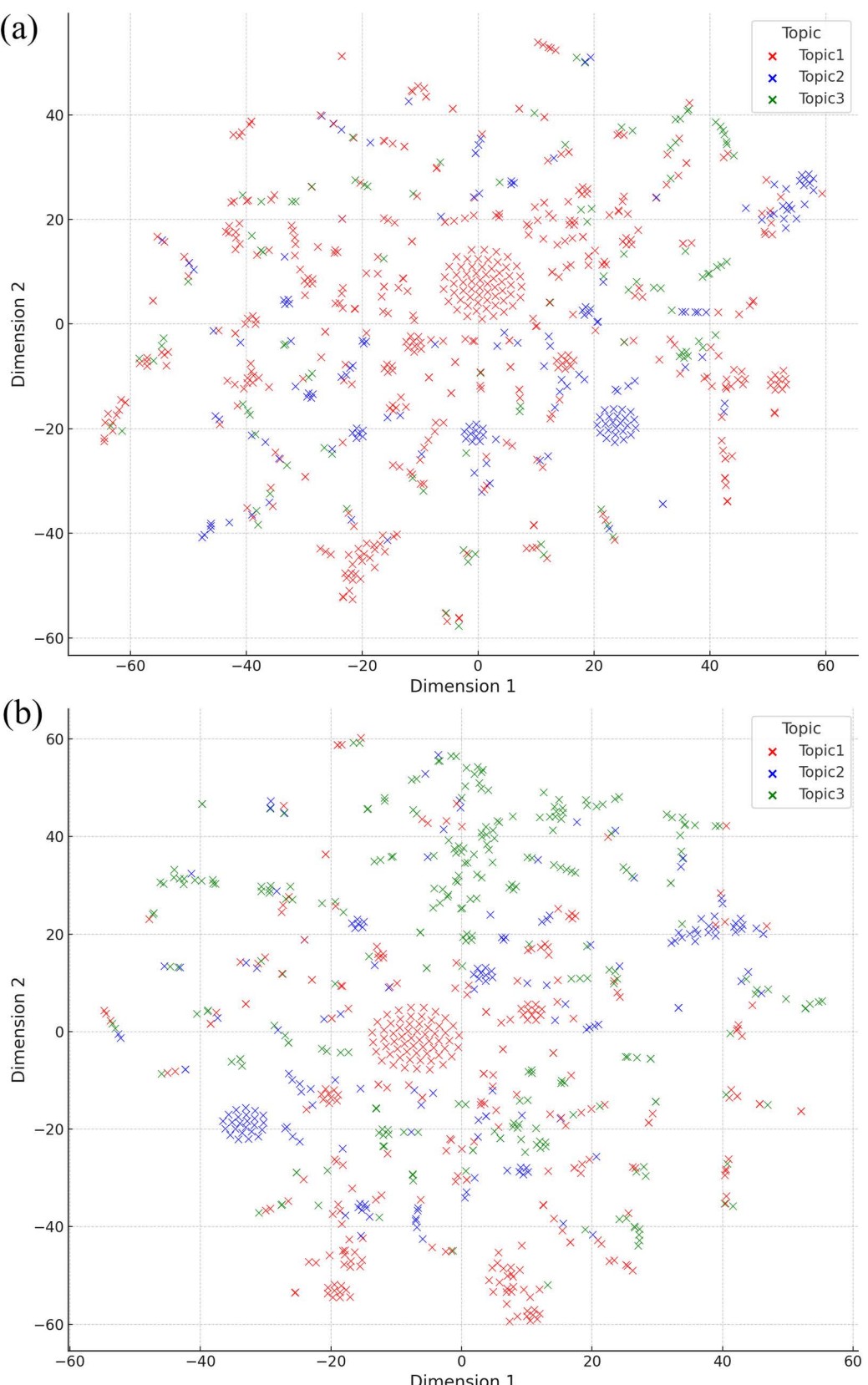

**Fig 6. t-SNE visualizations of TF-IDF matrix (a) before and (b) after concept decomposition.**

topic modeling faces limitations in accurately extracting biomedical themes related to different semantic types (e.g., symptoms, diseases, syndromes, mental conditions, and body parts, organs, or organ components) from social media texts. Our analysis shows that utilizing UMLS concept recognition through MetaMap can be more effective in identifying specific biomedical information within selected semantic types from social media texts by enabling a more precise clustering of highly associated medical concepts into relevant topics. By employing this strategy, redundant and irrelevant phrases are filtered out from the text corpus, establishing a valuable connection between self-reported health-related discussion data and phenotype characterization. The selection of UMLS Source Vocabularies in Meta-Map, such as ICD-9, ICD-10, and RxNorm, further contributes to this linkage. Importantly, these results also demonstrate that concept decomposition based on SNOMED CT relationships can be effective in biomedical information extraction. Before concept decomposition, medical concepts with parent-child relationships based on SNOMED precoordinated expressions are underlying in the preprocessed documents. The limited number of submissions and occurrences of certain concepts necessitate their separation into different topics through decomposition. Therefore, concept decomposition can amplify the occurrence of specified parent medical concepts while eliminating redundant child medical concepts across documents. This objective was evidenced by the significant improvement in topic modeling performance, including topic clustering, topic probability distribution, and topic-document relationships, with UMLS concept decomposition applied in this study. Importantly, from our expert reviews (section III, S2 Text), most experts agreed that the topics, once clustered following concept decomposition, could provide more interpretable and distinct clinical insights. Overall, the combined impact of the two strategies based on semantic relationships proves highly effective in uncovering biomedical information from online health-related discussions. This success highlights the two strategies as essential components of a novel clinical NLP pipeline.

This study further verifies the value of social media for mining health information by assessing the effects associated with statin use and the health conditions of online metabolic syndrome patients as a test case. First, based on the proportionality of topics, topic 3 is a prominently discussed side effect for statin-use patients, which is consistent with previous clinical findings [38,39]. We also found that patients receiving statin treatment who have muscle pain might also experience fatigue, tiredness, or weakness, concordant with prior studies [38,40,41]. Second, in addition to side effects, we found that metabolic syndrome patients prescribed statins were likely to discuss high cardiovascular risk. Indeed, based on the UMLS concepts and associated documents under topic 1, a majority of patients with metabolic syndrome reported a family history of heart attacks. Despite many patients expressing concerns and worries about the potential side effects of taking statins, they continued to inquire and demonstrate a strong need for statins as preventive care to mitigate their risk of myocardial infarction or stroke, consistent with long-standing research [42–44].

Along with these widely recognized topics associated with statin use, we also noted novel findings relative to topic 2: many metabolic syndrome patients in this online community exhibited significant mental health conditions, such as anxiety, depression, and nervousness. First, from associated submissions on topic 2, many patients prescribed statin treatment were anxious and depressed about their health conditions, including lab results (e.g., high cholesterol, high triglycerides, or high low-density lipoprotein), family history of metabolic syndrome, associated cardiovascular diseases, risk of heart attack and other heart diseases, as well as side effects of statins. Second, some patients in this online community also developed anxiety and apprehension related to taking medicine. Given the possibility of multiple

diagnoses (including related comorbidities), some patients were prescribed additional medications (beyond statins) in different or the same periods (e.g., evolocumab, fenofibrate). Consequently, some patients expressed concern about potential interactions or the lack of an optimized treatment plan. Additionally, we found that some patients reported the onset of anxiety, depression, and mood swings after the initiation of statins. However, this finding contradicts current research on the effects of statins, as most studies and mechanistic evidence suggest an antidepressant effect for statins [45,46]. Notably, some patients in this online community mentioned that when they described experiences of anxiety, depression, and nervousness to their physicians and suggested these feelings might be side effects of taking statins, physicians commonly remained unconvinced. As further demonstration of heightened concerns about anxiety and depression among statin users, our EHR validation in two real-world clinical databases indicated that statin users had a higher prevalence of depression or anxiety compared to matched non-users, suggesting the importance of addressing mental health issues such as anxiety and depression to physicians caring for patients taking statins. Therefore, this study reveals that mental health issues related to anxiety and depression are common symptoms among social media posts from statin users. The finding highlights the need for healthcare providers to actively monitor and address mental health symptoms, such as anxiety and depression, in patients using statins. They may consider integrating routine mental health screenings into follow-up visits for statin users and providing resources or referrals for mental health support when necessary.

## Limitations and future work

This paper has certain limitations. First, our focus was solely on the statin use within a single online community, and thus, the findings may not be generalizable to all other online communities. Second, due to the absence of demographic data for the patients in this online community, we could not conduct a comprehensive population analysis. Third, although we proposed techniques to enhance UMLS concept recognition's effectiveness, MetaMap may still incorrectly identify some UMLS medical concepts, potentially impacting topic modeling performance. Finally, in our EHR validation studies we did not consider the timing of anxiety/depression diagnoses relative to statin initiation; therefore, while our results suggest that statin exposure is associated with anxiety and depression, we cannot definitively identify these conditions as side effects of statin use. In the future, we intend to expand our research by studying additional online communities for statin users and collecting demographic information about their users through a survey questionnaire, aiming to generalize our findings. We also intend to explore the applications of large language models to identify the medical concepts from social media texts. In addition, further research is needed to clarify the temporal relationship between statin use and anxiety or depression by integrating this temporal aspect into the EHR validation.

## Conclusion

Broadly, our study demonstrated refined approaches for gathering patient-reported drug experiences, health concerns, and mental conditions from social media. These findings further confirm the potential of social media as a valuable resource for medical research. This new topic modeling approach, as used for mining statin use from online self-reported health discussions, could be extended to investigate other conditions or medications on social media, highlighting the uniqueness of social media as a source of self-reported health information. These results underscore the importance of leveraging social media for real-world insights into patient perspectives and treatment outcomes.

## Supporting information

**S1 Text.  Phecodes and ICD codes for anxiety/depression.**
(DOCX)

**S2 Text.  Appendix.**
(DOCX)

**S3 Text.  Example submissions for each topic in the final STM.**
(DOCX)

## Acknowledgments

The authors would like to acknowledge the NIH All of Us Research Program and VUMC BioVU Program.

## Author contributions

**Conceptualization:** Yi Xin, Zhijun Yin, Wei-Qi Wei.

**Data curation:** Yi Xin.

**Funding acquisition:** Wei-Qi Wei.

**Methodology:** Yi Xin, Monika E. Grabowska, Srushti Gangireddy, Alyson L. Dickson, Qiping Feng, Zhijun Yin, Wei-Qi Wei.

**Software:** Yi Xin.

**Supervision:** Wei-Qi Wei.

**Validation:** Yi Xin, Monika E. Grabowska, Matthew S. Krantz, V. Eric Kerchberger, Wei-Qi Wei.

**Visualization:** Yi Xin.

**Writing – original draft:** Yi Xin.

**Writing – review & editing:** Yi Xin, Monika E. Grabowska, Alyson L. Dickson, Qiping Feng, Zhijun Yin, Wei-Qi Wei.

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
