## [Decision Letter · Decision Letter 0]

28 Oct 2024

PONE-D-24-41279Improving Topic Modeling Performance on Social Media Through Semantic Relationships within Biomedical TerminologyPLOS ONE

Dear Dr. Wei,

Thank you for submitting your manuscript to PLOS ONE. After careful consideration, we feel that it has merit but does not fully meet PLOS ONE’s publication criteria as it currently stands. Therefore, we invite you to submit a revised version of the manuscript that addresses the points raised during the review process.

We look forward to receiving your revised manuscript.

Kind regards,

Zhe He, PhD

Academic Editor

PLOS ONE

Journal Requirements:

3. We note that you have indicated that there are restrictions to data sharing for this study. PLOS only allows data to be available upon request if there are legal or ethical restrictions on sharing data publicly. For more information on unacceptable data access restrictions, please see http://journals.plos.org/plosone/s/data-availability#loc-unacceptable-data-access-restrictions. Before we proceed with your manuscript, please address the following prompts: a) If there are ethical or legal restrictions on sharing a de-identified data set, please explain them in detail (e.g., data contain potentially identifying or sensitive patient information, data are owned by a third-party organization, etc.) and who has imposed them (e.g., a Research Ethics Committee or Institutional Review Board, etc.). Please also provide contact information for a data access committee, ethics committee, or other institutional body to which data requests may be sent. b) If there are no restrictions, please upload the minimal anonymized data set necessary to replicate your study findings to a stable, public repository and provide us with the relevant URLs, DOIs, or accession numbers. For a list of recommended repositories, please see https://journals.plos.org/plosone/s/recommended-repositories. You also have the option of uploading the data as Supporting Information files, but we would recommend depositing data directly to a data repository if possible. We will update your Data Availability statement on your behalf to reflect the information you provide.

Additional Editor Comments:

The reviewers raised a number of issues including comparing with traditional topic modeling approaches, the rationale of structural topic modeling, and various technical details. Please address these issues in the revision.

Reviewers' comments:

Reviewer's Responses to Questions

**Comments to the Author**

1. Is the manuscript technically sound, and do the data support the conclusions?

Reviewer #1: Yes

Reviewer #2: No

Reviewer #3: Partly

2. Has the statistical analysis been performed appropriately and rigorously? 

Reviewer #1: Yes

Reviewer #2: No

Reviewer #3: Yes

3. Have the authors made all data underlying the findings in their manuscript fully available?

Reviewer #1: Yes

Reviewer #2: Yes

Reviewer #3: Yes

4. Is the manuscript presented in an intelligible fashion and written in standard English?

Reviewer #1: Yes

Reviewer #2: Yes

Reviewer #3: Yes

5. Review Comments to the Author

Reviewer #1: Summary:

This paper presents a novel approach to improving topic modeling performance on social media data in the healthcare domain. The authors introduce a semantic-type-based topic modeling pipeline that integrates UMLS concept recognition and SNOMED CT relationship-based concept decomposition into traditional topic modeling. They demonstrate the effectiveness of this approach using discussions about statin use from Reddit as a case study. The main contributions include:

1. Enhancing topic modeling by leveraging semantic relationships within biomedical terminology.

2. Improving the extraction of health-related themes from noisy social media data.

3. Validating findings using real-world clinical data from electronic health records.

The innovation lies in the combination of UMLS concept recognition, SNOMED CT relationship-based concept decomposition, and traditional topic modeling to create a more robust method for analyzing health-related social media content.

Specifics

While the introduction does mention some related work (e.g., lines 36-52 discuss various applications of topic modeling in healthcare), the authors don't explicitly compare their approach to these existing methods in terms of performance or outcomes.

Some potential comparisons could include:

a) Traditional topic modeling approaches (e.g., Latent Dirichlet Allocation) on the same dataset without semantic enrichment.

b) Other semantic-enhanced topic modeling methods that have been applied to healthcare social media data.

c) Alternative approaches to extracting health-related information from social media, such as supervised machine learning methods or rule-based systems.

Introduction:

- Line 36: "Topic modeling is an unsupervised machine learning technique in natural language processing (NLP) used" - Consider rephrasing to improve flow, e.g., "Topic modeling, an unsupervised machine learning technique in natural language processing (NLP), is used"

Methods:

- Line 84: "We conducted web scraping of a repository containing dump files to collect the submissions and comments" - Consider providing more details on the ethical considerations of data collection from social media

Results:

- Figure 2: Consider adding a brief explanation of how to interpret the topic proportions in the figure caption

Discussion:

- Line 315: "Along with these widely recognized topics associated with statin use, we also noted novel findings relative to topic 2:" - Consider elaborating on the potential implications of these novel findings for healthcare providers and researchers

Limitations and future work:

- Consider expanding on potential solutions or approaches to address the limitations mentioned

- Some sentences are quite long and complex. Consider breaking them down for improved readability

- Ensure consistent capitalization of terms throughout the paper (e.g., "Topic Modeling" vs. "topic modeling")

- Consider adding a brief section on the ethical considerations of using social media data for healthcare research

Reviewer #2: The paper presents a method for improving the extraction of meaningful health-related themes from social media, which could lead to better patient insights and enhanced clinical decision-making.

Comments:

1) The rationale behind using STM is not clear, as STM is used to identify the temporal variation and difference in source data. As far as temporal evidence how did the topics change over time period is not specified.

2) The Figures provided are not clear, a pipeline/ algorithm describing the entire flow would add value to the manuscript.

3) How do you address the 329 documents which did not have a matching UMLS concepts from the restricted list.

4) How is the value of K between 2-6 selected, what is the maximum number of topics generated by the model.

5) Line 124 the authors explain the decomposition of concepts to parent concepts, what is the criteria for identifying the concepts to be decomposed, please explain the logic behind this step.

6) In the EHR validation study association between Satin exposure and Mental Health was performed, even though the Subreddit was cholesterol. Why the association study of topic 2 was selected, please validate the reasoning for deciding on the EHR validation study.

7) More examples of topics generated and for each topic did the results had metadata text attached to it, please clarify with execution output.

Reviewer #3: This study effectively demonstrates the integration of UMLS concept recognition and concept decomposition, based on SNOMED CT relationships, into traditional topic modeling frameworks to enhance the identification of meaningful biomedical topics. The manuscript is well-structured and clearly written, making the methodology and findings accessible. However, I have the following suggestions for the authors’ consideration that could further improve the overall clarity and impact of the manuscript.

1. In the Background section, the authors discuss the limitations of traditional topic modeling. However, in the Methods section, they focus on Structural Topic Modeling (STM). It would enhance clarity to explicitly define the gap between STM and existing methods, reinforcing why STM is a better choice for this analysis.

2. As there are several topic modeling approaches available, including STM and LDA, the authors should provide a clear rationale for selecting Structural Topic Modeling (STM) over the more widely used Latent Dirichlet Allocation (LDA). This would make the choice of approach more convincing, particularly in terms of how STM enhances the research in comparison to other methodologies.

3. Since this study uses unsupervised approaches, it is important to explain the selection of k (the number of topics) and the initialization process for clustering. The selection of k can greatly influence the results, and more detail here would enhance the paper’s rigor.

4. In the UMLS Concept Recognition section, the authors mention four semantic types. It would strengthen the paper to explain why these specific types were chosen and how they align with the context of the analysis.

5. In the STM Model Setup section, providing more information on how the number of topics (k) was chosen, and elaborating on the qualitative analysis performed to assess the top words in each topic would improve transparency.

6. In the Concept Decomposition Based on SNOMED CT Relationships section, it would be beneficial to include an example to demonstrate the concept decomposition process. This will help clarify how granularity issues with SNOMED CT were handled.

7. The process of identifying pre-coordinated expressions is unclear. Explaining whether this was done manually or automatically, and outlining the identification method would enhance understanding. The author cited a JAMIA paper “The use of SNOMED CT, 2013-2020: a literature review.” however, in this paper it said “no evaluation was made as to whether the post-coordinated expression improved the performance of normalization tasks compared to only pre-coordination coding schemes.” it did not indicate the process of pre-coordinate concept identification. Could the authors please explain how you identify pre-coordinated expression?

9. The decomposition methodology needs further illustration, especially regarding how it handles pre-coordinated concepts. Without more information, it is difficult to understand how this process works in practice.

10. If the method is not intended to be generalized, this should be explicitly addressed in the Discussion section, especially given the complexity of handling a large list of concepts.

11. Clarifying terms like "word-topic distribution" early in the paper would make the content more accessible to a broader audience.

12. The comparison of concept decomposition in the STM model is presented without sufficient explanation. If concept decomposition is a key step in improving the topic modeling, this needs to be made more explicit in the manuscript.

6. PLOS authors have the option to publish the peer review history of their article (what does this mean? ). If published, this will include your full peer review and any attached files.

**Do you want your identity to be public for this peer review?** For information about this choice, including consent withdrawal, please see our Privacy Policy .

Reviewer #1: No

Reviewer #2: No

Reviewer #3: No

---

## [Author Response · Author response to Decision Letter 0]

12 Dec 2024

Response to Reviewers

Thank you for the opportunity to respond to comments received for our application, “Improving Topic Modeling Performance on Social Media Through Semantic Relationships within Biomedical Terminology”. We appreciate the committee’s suggestions and positive comments including: “the innovation lies in the combination of UMLS concept recognition, SNOMED CT relationship-based concept decomposition, and traditional topic modeling to create a more robust method for analyzing health-related social media content”; “the paper presents a method for improving the extraction of meaningful health-related themes from social media, which could lead to better patient insights and enhanced clinical decision-making”; “the manuscript is well-structured and clearly written, making the methodology and findings accessible”; etc.

Below, we address the reviewers’ concerns (the line numbers mentioned in this letter are from Revised Manuscript with Track Changes):

Reviewer 1:

Q1: While the introduction does mention some related work (e.g., lines 36-52 discuss various applications of topic modeling in healthcare), the authors don't explicitly compare their approach to these existing methods in terms of performance or outcomes.

Some potential comparisons could include:

a) Traditional topic modeling approaches (e.g., Latent Dirichlet Allocation) on the same dataset without semantic enrichment.

b) Other semantic-enhanced topic modeling methods that have been applied to healthcare social media data.

c) Alternative approaches to extracting health-related information from social media, such as supervised machine learning methods or rule-based systems.

Answer: Thanks for the comments. We have expanded the Introduction to incorporate greater rationale for the choice of STM in introduction section (lines 60-65): “The structural topic model (STM) is one of the most common types of topic modeling approaches used widely in analysis of healthcare data and social media health discussions, as it excels in uncovering latent themes and temporal trends. Unlike other topic modeling approaches, such as Latent Dirichlet Allocation, STM is uniquely able to incorporate document-level metadata, such as user attributes, simultaneously with topic inference, making it particularly advantageous for analyzing social media data in health-related NLP tasks.” In our study, the document-level metadata includes submission creation time, comment count, score for each submission. This integration enhances the interpretability and relevance of the extracted topics. For instance, the STM allows researchers to estimate topic models that include document-level metadata, providing rich ways to explore topics, estimate uncertainty, and visualize quantities of interest [1,2]. In addition, although LDA was popular in many text mining studies, it has been well documented in the empirical literature that LDA is sub-optimal for short texts published in social media and needs to be improved to analyze noisy social media texts [3]. Therefore, for (a): we think the STM applied in our study should perform better than the traditional LDA in our dataset. In addition, our focus is to develop a new semantic-type-based topic modeling approach to cluster more meaningful topics in clinical perspective. Hence, the application of other traditional approaches (e.g., LDA) is not necessary and/or may provide less productive results for this dataset. For (b), our semantic-type-based topic modeling approach generates topics comprised of standard medical concepts, making them more interpretable for clinicians and doctors. This is one of our study’s merits. To date, there does not exist any other semantic-based topic modeling approach that can generate topics comprised of standard medical concepts, whether it has been applied to healthcare social media data or not. For (c), supervised approaches, such as Support Vector Machines (SVMs) and neural networks, are trained on labeled datasets to classify text into predefined topics. Rule-based systems utilize predefined linguistic rules and keyword matching to identify topics within the text. However, these methods have certain limitations compared to unsupervised topic modeling approaches. Supervised methods require extensive labeled data, which can be labor-intensive to produce and may not capture the full range of topics present in dynamic social media content. Rule-based systems, while straightforward, often lack the flexibility to adapt to the evolving language and slang typical of social media platforms. In contrast, topic modeling approaches is more adaptable to the nuances of social media language [3].

Q2:

Introduction:

- Line 36: "Topic modeling is an unsupervised machine learning technique in natural language processing (NLP) used" - Consider rephrasing to improve flow, e.g., "Topic modeling, an unsupervised machine learning technique in natural language processing (NLP), is used"

Answer: Thanks for the suggestion. We have revised this sentence accordingly.

Q3:

Methods:

Line 84: "We conducted web scraping of a repository containing dump files to collect the submissions and comments" - Consider providing more details on the ethical considerations of data collection from social media

Answer: Thanks for the suggestion. The ethical considerations of using social media data for healthcare research was added in the Methods section (lines 117-120): “In terms of ethical considerations of using social media data for healthcare research, we ensured adherence to ethical standards and for data collection from social media platforms. Specifically, the extracted social media data was de-identified and no personal characteristics from users were collected.”

Q4:

Results:

Figure 2: Consider adding a brief explanation of how to interpret the topic proportions in the figure caption

Answer: Thanks for the suggestion. The brief explanation of the topic proportions has been add to lines 213-214: “Topic proportion is the percentage of the documents associated with a specific topic, indicating its prevalence”.

Q5:

Discussion:

Line 315: "Along with these widely recognized topics associated with statin use, we also noted novel findings relative to topic 2:" - Consider elaborating on the potential implications of these novel findings for healthcare providers and researchers

Answer: Thanks for the suggestion. The potential implications of these novel findings has been added to lines 378-381: “The finding highlights the need for healthcare providers to actively monitor and address mental health symptoms, such as anxiety and depression, in patients using statins. They may consider integrating routine mental health screenings into follow-up visits for statin users and providing resources or referrals for mental health support when necessary.”.

Q6:

Limitations and future work:

- Consider expanding on potential solutions or approaches to address the limitations mentioned

Answer: Thanks for the suggestion. The potential solutions or approaches has been added to lines 391-397: “In the future, we intend to expand our research by studying additional online communities for statin users and collecting demographic information about their users through a survey questionnaire, aiming to generalize our findings. We also intend to explore the applications of large language models to identify the medical concepts from social media texts. In addition, further research is needed to clarify the temporal relationship between statin use and anxiety or depression by integrating this temporal aspect into the EHR validation.”.

Q7:

-sentences are quite long and complex. Consider breaking them down for improved readability

- Ensure consistent capitalization of terms throughout the paper (e.g., "Topic Modeling" vs. "topic modeling")

- Consider adding a brief section on the ethical considerations of using social media data for healthcare research

Answer: We have revised our manuscript to address the above comments. Particularly, the ethical considerations of using social media data for healthcare research was added in the Methods section (lines 117-120).

Reviewer 2:

Q1: The rationale behind using STM is not clear, as STM is used to identify the temporal variation and difference in source data. As far as temporal evidence how did the topics change over time period is not specified.

Answer: Thanks for the comment. We appreciate your observation regarding the typical application of STM for analyzing temporal variation in social media data. Our rationale for employing STM in this study extends beyond its temporal capabilities. We used STM primarily for its ability to incorporate metadata into topic inference, such as user attributes. Therefore, the primary aim of our analysis is to understand the topics related to statin use from clinical perspective, rather than the evolution of topics over time. This is because patients with metabolic syndromes and long-term statin use typically exhibit symptoms and side effects that are relatively stable over time, since metabolic syndrome is a chronic condition characterized by persistent factors such as high blood pressure, insulin resistance, dyslipidemia, and central obesity. We have provided additional rationale for the selection of STM in the Introduction (see response to Reviewer 1, Q1).

Q2: The Figures provided are not clear, a pipeline/ algorithm describing the entire flow would add value to the manuscript.

Answer:

Thanks for the suggestion. We have added a pipeline overview in Methods section (lines 108-114): “In this study, we began by collecting and preprocessing social media data extracted from Reddit. We then conducted an initial round of STM using a traditional STM pipeline. Next, we enhanced this pipeline by incorporating semantic type information through UMLS concept recognition using MetaMap and performed a second round of STM. Subsequently, we further extended the pipeline by integrating SNOMED precoordinated expressions through concept decomposition, followed by a final round of STM. We then compared the results from the second and final STM rounds. Finally, we validated a newly identified topic from the final round of STM to demonstrate the effectiveness of our approach.”.

Q3: How do you address the 329 documents which did not have a matching UMLS concepts from the restricted list.

Answer: Thanks for the question. We excluded documents without matching UMLS concepts from the restricted list during the topic modeling process. While we acknowledge that the number of such documents is notable, the majority contained posts where patients primarily discussed their lab test results or sought general advice, with no relevance to our targeted semantic types. This lack of relevance reflects the inherently noisy nature of discussions in this online community, which often include content not directly aligned with the focus of our analysis for understanding the sides effects and concerns of statin users.

Q4: How is the value of K between 2-6 selected, what is the maximum number of topics generated by the model.

Answer: Thanks for the question. The choice of K (2–6) was based on the nature of the data and experimental evaluations. Discussions among statin users in online communities typically focus on a limited set of symptoms and side effects, so a small number of topics aligns with the content's scope. We tested various K values and found that 2–6 produced coherent and interpretable topics, while larger values led to fragmentation. The model itself does not impose a maximum number of topics; K is defined by the user based on the data and application.

Q5: Line 124 the authors explain the decomposition of concepts to parent concepts, what is the criteria for identifying the concepts to be decomposed, please explain the logic behind this step.

Answer: We described the criteria for identifying the concepts to be decomposed in the Concept Decomposition Based on SNOMED CT Relationship section: namely, we first identified UMLS concepts with significant word-topic distribution β values (β > 0.02) in each topic and compiled these into a list. Then, we searched each of the UMLS concepts from this list in SNOMED CT and examined whether it had precoordinated expressions. The logic behind the step is that we would like to identify those high-frequency concepts which have parent concepts based on SNOMED relationship and decompose those identified concepts to get a more distinct and meaningful clustering of topics. This ensures semantic clarity while reducing redundancy.

Q6: In the EHR validation study association between Statin exposure and Mental Health was performed, even though the Subreddit was cholesterol. Why the association study of topic 2 was selected, please validate the reasoning for deciding on the EHR validation study.

Answer: Thanks for the question. We identified three topics: topic 1 (“metabolic syndrome”), topic 2 (“anxiety and depression”), and topic 3 (“muscle pain”). While metabolic syndrome (e.g., hypercholesterolemia) and muscle pain are well-known among statin users, the relationship between statin use and mental health is less explored. To address this gap, we selected topic 2 for validation in the EHR database to assess its relevance and legitimacy.

Q7: More examples of topics generated and for each topic did the results had metadata text attached to it, please clarify with execution output.

Answer:

Thank you for your comment. Using STM, we generated topics based on the word-topic distribution, which provides the top words associated with each topic. While the model itself does not directly attach metadata text to topics, we can link the topics to metadata by examining the document-topic associations. To address your request, we include examples of the metadata text associated with each topic in the supplementary (S3 text) for further clarity.

Reviewer 3:

Q1: In the Background section, the authors discuss the limitations of traditional topic modeling. However, in the Methods section, they focus on Structural Topic Modeling (STM). It would enhance clarity to explicitly define the gap between STM and existing methods, reinforcing why STM is a better choice for this analysis.

Answer:

Thanks for the comment. We agree that explicitly defining the gap between STM and traditional topic modeling approaches would enhance clarity, and we have revised the descriptions for this in the Introduction section. Unlike other traditional methods, which often struggle with the noisy, dynamic nature of social media data and lack mechanisms to incorporate rich contextual information, STM is specifically designed to address these challenges [1,2,4]. STM allows the integration of document-level metadata, such as user attributes and temporal information, enabling it to effectively uncover latent themes and temporal trends in social media health discussions. Given these strengths, STM is a more robust choice for our analysis, and we consider it well-suited as the baseline approach in our study.

Q2: As there are several topic modeling approaches available, including STM and LDA, the authors should provide a clear rationale for selecting Structural Topic Modeling (STM) over the more widely used Latent Dirichlet Allocation (LDA). This would make the choice of approach more convincing, particularly in terms of how STM enhances the research in comparison to other methodologies.

Answer:

Thanks for the comment. We have provided additional rationale for the selection of STM (see response to Reviewer 1, Q1). The STM offers advantages over Latent Dirichlet Allocation (LDA) when analyzing social media data, primarily due to its ability to incorporate document-level metadata, such as user attributes. In our study, the document-level metadata includes submission creation time, comment count, score for each submission. This integration enhances the interpretability and relevance of the extracted topics. For instance, the STM allows researchers to estimate topic models that include document-level metadata, providing rich ways to explore topics, estimate uncertainty, and visualize quantities of interest [1,2]. Also, although LDA was popular in many text mining studies, it has been well documented

---

## [Decision Letter · Decision Letter 1]

21 Jan 2025

Improving Topic Modeling Performance on Social Media Through Semantic Relationships within Biomedical Terminology

PONE-D-24-41279R1

Dear Dr. Wei,

We’re pleased to inform you that your manuscript has been judged scientifically suitable for publication and will be formally accepted for publication once it meets all outstanding technical requirements.

Kind regards,

Zhe He, PhD

Academic Editor

PLOS ONE

Additional Editor Comments (optional):

The authors have adequately addressed the reviewers' comments.

Reviewers' comments:

Reviewer's Responses to Questions

**Comments to the Author**

1. If the authors have adequately addressed your comments raised in a previous round of review and you feel that this manuscript is now acceptable for publication, you may indicate that here to bypass the “Comments to the Author” section, enter your conflict of interest statement in the “Confidential to Editor” section, and submit your "Accept" recommendation.

Reviewer #1: All comments have been addressed

Reviewer #3: All comments have been addressed

2. Is the manuscript technically sound, and do the data support the conclusions?

Reviewer #1: Yes

Reviewer #3: Yes

3. Has the statistical analysis been performed appropriately and rigorously? 

Reviewer #1: Yes

Reviewer #3: N/A

4. Have the authors made all data underlying the findings in their manuscript fully available?

Reviewer #1: Yes

Reviewer #3: Yes

5. Is the manuscript presented in an intelligible fashion and written in standard English?

Reviewer #1: Yes

Reviewer #3: Yes

6. Review Comments to the Author

Reviewer #1: (No Response)

Reviewer #3: Thanks to the authors for thoroughly addressing my previous comments. I have reviewed the revised manuscript and the authors' detailed responses to the review feedback. I believe the authors have adequately addressed my concerns. I appreciate their thoughtful consideration and the effort they have put into the revisions.

7. PLOS authors have the option to publish the peer review history of their article (what does this mean? ). If published, this will include your full peer review and any attached files.

**Do you want your identity to be public for this peer review?** For information about this choice, including consent withdrawal, please see our Privacy Policy .

Reviewer #1: No

Reviewer #3: No

---

## [Editor Report · Acceptance letter]

PONE-D-24-41279R1

PLOS ONE

Dear Dr. Wei,

I'm pleased to inform you that your manuscript has been deemed suitable for publication in PLOS ONE. Congratulations! Your manuscript is now being handed over to our production team.

Kind regards,

on behalf of

Dr. Zhe He

Academic Editor

PLOS ONE